Methods

# Simultaneous multiple allelic replacement in the malaria parasite enables dissection of PKG function

Konstantinos Koussis[1] , Chrislaine Withers-Martinez[1], David A Baker[2], Michael J Blackman[1,2]

**Over recent years, a plethora of new genetic tools has transformed conditional engineering of the malaria parasite genome, allowing functional dissection of essential genes in the asexual and sexual blood stages that cause pathology or are required for disease transmission, respectively. Important challenges remain, including the desirability to complement conditional mutants with a correctly regulated second gene copy to confirm that observed phenotypes are due solely to loss of gene function and to analyse structure–function relationships. To meet this challenge, here we combine the dimerisable Cre (DiCre) system with the use of multiple *lox* sites to simultaneously generate multiple recombination events of the same gene. We focused on the *Plasmodium falciparum* cGMP-dependent protein kinase (PKG), creating in parallel conditional disruption of the gene plus up to two allelic replacements. We use the approach to demonstrate that PKG has no scaffolding or adaptor role in intraerythrocytic development, acting solely at merozoite egress. We also show that a phosphorylation-deficient PKG is functionally incompetent. Our method provides valuable new tools for analysis of gene function in the malaria parasite.**

## Introduction

From the early documentation of targeted gene disruption in yeast by homologous recombination (1) to the use of site-specific recombinases (2) and the development of gene-editing tools such as CRISPR (3, 4), the ability to modify DNA has revolutionised understanding of gene function in model organisms and pathogens. *Plasmodium* spp., the protozoan parasites that are the aetiological agents of malaria, are responsible for more than 400,000 deaths per year (5), with *Plasmodium falciparum* causing the deadliest form of the disease. Widespread resistance to frontline antimalarial drugs and the absence of an effective vaccine make the identification of new antimalarial drug targets a necessity (6), but to achieve this, an improved understanding of the biology of

the parasite is required. Transient transfection of *Plasmodium* was first reported almost 3 decades ago (7), but in part due to the haploid genome of the parasite, functional studies of essential genes in the asexual blood stages that are responsible for all the clinical manifestations of the disease have been extremely difficult.

Conditional deletion or rearrangement of DNA segments through activation of site-specific recombinases such as Cre has been the gold-standard system for gene editing in many model organisms, but attempts to adapt the Cre-*lox* system to blood stages of *P. falciparum* initially failed because of difficulties in suppressing constitutive activity of the recombinase (8, 9). This problem was solved with the adaptation of the dimerisable Cre (DiCre) system initially for *Toxoplasma* and subsequently for *P. falciparum* blood stages (10, 11). In this approach, Cre is expressed in the form of two enzymatically inactive domains, each of which is fused to a small rapamycin-binding protein. In the presence of rapamycin (RAP), the two fusion proteins heterodimerise, rapidly inducing Cre activity (12, 13, 14, 15). The versatility of this system in *Plasmodium* was substantially enhanced with the development of the *loxPint* module, which elegantly allowed intragenic introduction of "silent" *loxP* sites within a small synthetic intron (16). Over the past 5 yr, additional modifications have been made to the *P. falciparum* DiCre system, including installation of the DiCre cassette into alternative chromosomal loci and use of different *P. falciparum* strains, and the approach has now been exploited for the functional analysis of many essential genes (17, 18, 19, 20, 21, 22, 23, 24, 25, 26, 27, 28, 29). Very recently, DiCre has been adapted to other *Plasmodium* species, including the widely-used rodent malaria model *Plasmodium berghei* and the zoonotic pathogen *Plasmodium knowlesi* (30, 31). However, application of the system for the simultaneous generation of both loss-of-function and genetically complemented parasite lines has remained technically challenging.

Signalling through cyclic 3′,5′-guanosine monophosphate (cGMP) plays important roles in many eukaryotes, including the malaria parasite. The only known sensor of cGMP signalling in the parasite is its cGMP-dependent protein kinase (PKG), which is encoded by a single-copy gene in all *Plasmodium* species (32, 33). Chemical genetic and genetic approaches have shown that cGMP

[1]Malaria Biochemistry Laboratory, Francis Crick Institute, London, UK    [2]Faculty of Infectious and Tropical Diseases, London School of Hygiene & Tropical Medicine, London, UK

Correspondence: konstantinos.kousis@crick.ac.uk; mike.blackman@crick.ac.uk

signalling is essential for key developmental transitions throughout the entire parasite life cycle, including activation of sexual forms (gametogenesis), ookinete formation and motility in the mosquito vector, sporozoite motility and liver cell infection, and maturation and release (egress) of liver-stage merozoites into the bloodstream of the vertebrate host (34, 35, 36, 37, 38, 39, 40). In the asexual blood stages of *P. falciparum*, the use of highly selective small-molecule inhibitors of PKG, combined with the use of gatekeeper muta- genesis to confirm on-target selectivity of the compounds, has convincingly demonstrated that PKG catalytic activity is not re- quired during the ~48-h-long period of intraerythrocytic parasite development (schizogony) but has an essential role in asexual blood-stage schizont rupture and merozoite egress (37, 38, 41). More recently, conditional down-regulation of PKG expression using a destabilisation domain approach confirmed that the kinase is required for blood-stage viability; however, in this study, the specific point(s) in the erythrocytic cycle at which proliferation arrested was not documented (42). Many kinases and other sig- nalling proteins operate within multicomponent signalling com- plexes in which individual members play critical scaffolding or adaptor roles in addition to any canonical enzymatic roles (43, 44, 45, 46, 47). The various studies alluded to above have clearly indicated that PKG enzyme activity is not required for intra- erythrocytic parasite development. However, whether PKG plays additional scaffolding role(s) throughout intraerythrocytic de- velopment that may not require catalytic activity is unknown. Many other aspects of PKG biology are also still unexplored, including the importance of PKG phosphorylation, a modification that often plays a role in kinase activation and function (48). Several global phosphoproteomic studies have identified a total of seven residues that are phosphorylated in PKG; two of these lie within a cyclic nucleotide-binding domain and five within the kinase domain (49, 50, 51, 52, 53). Whether phosphorylation of these residues is im- portant for PKG function has not been examined.

Here, we address these important questions. To enable this, we describe a modified DiCre system that takes advantage of sto- chastic Cre-mediated recombination between different *lox* sites (54), allowing us to perform in parallel conditional disruption and allelic replacement of the *P. falciparum* PKG gene, with the distinct events indicated by the expression of distinct fluorescent reporter proteins. Our new system provides valuable new tools for condi- tional genetics in *Plasmodium*. Using it, we genetically demonstrate for the first time that PKG has no essential scaffolding role during intraerythrocytic malaria parasite development, and we show that a phosphorylation-deficient form of PKG is nonfunctional.

# Results

## Introduction of two contiguous *lox* sites into a short intron to allow multiple simultaneous conditional gene modifications

We first re-designed the *loxPint* module (16) to incorporate two contiguous non-overlapping 34-bp sequences, *loxN* and *lox2272*. These modified *lox* sites have previously been shown to be in- compatible with *loxP* or with each other in different systems (54, 55). To validate the functionality of the modified *loxPint* (called *2lox- Pint*) and to assess its capacity to undergo correct splicing, we inserted the *2loxPint* element into the *P. falciparum* gene encoding the cGMP-dependent PKG. The *P. falciparum* PKG gene (*pfpkg*: PF3D7_1436600) comprises 4 introns and 5 exons, with all four of its consensus cyclic nucleotide-binding domains and the kinase do- main encoded by exons 3–5. Using a Cas9 expression plasmid (19) to mediate targeted double-stranded DNA cleavage and a marker-free rescue construct to enable repair by homologous recombination, we precisely replaced intron 3 of *pfpkg* with *2loxPint* in the DiCre- expressing *P. falciparum* B11 line (Fig 1A and B) (25). Limiting dilution cloning of the modified parasites generated a genetically homogenous

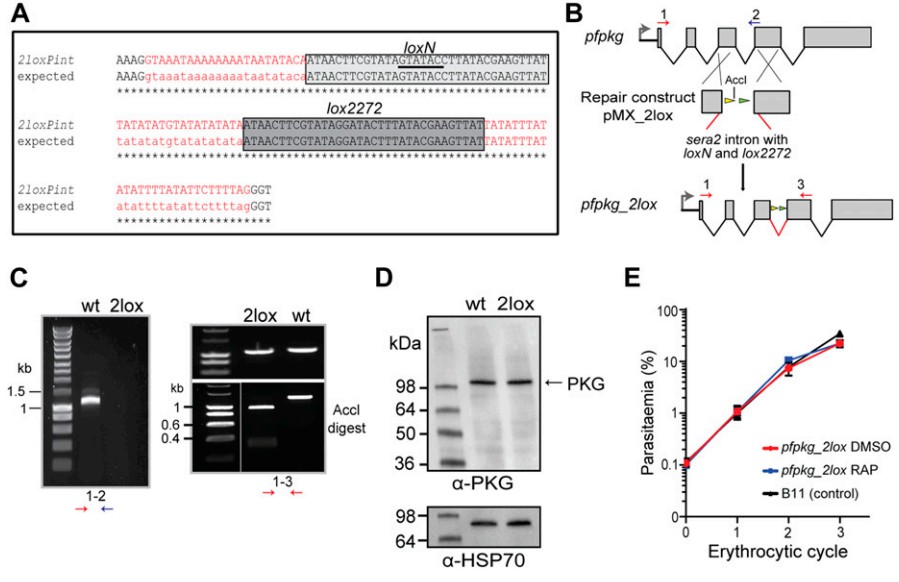

**Figure 1. Replacement of *pfpkg* intron 3 with *2loxPint* allows normal PKG expression and parasite replication.**
**(A)** Shown is the integrated *2loxPint* sequence derived by nucleotide sequencing of genomic DNA from the *P. falciparum pfpkg_2lox* line, aligned with the expected sequence (both in red). The *loxN* and *lox2272* sites are in grey boxes, with the unique internal AccI site underlined. Boundaries of the 3' end of exon 3 (AAAG) and the 5' end of exon 4 (GGT) are shown in black. **(B)** Schematic representation of the strategy used to replace intron 3 of the *pfpkg* gene with *2loxPint* in the DiCre-expressing B11 *P. falciparum* line. Positions of oligonucleotides used for diagnostic PCR are indicated (red and blue arrows). **(C)** (Left) Diagnostic PCR results showing absence of the endogenous intron 3 in the *pfpkg_2lox* line (2lox) relative to the parental B11 line (wt). (Right) Diagnostic PCR results and restriction digest of the PCR amplicon with AccI, showing the expected digestion only of the amplicon from the *pfpkg_2lox* line (2lox). **(D)** Western blot analysis of extracts of wt and *pfpkg_2lox* schizonts, showing similar expression levels of PKG (~98 kD, indicated). Antibodies to the cytoplasmic parasite protein HSP70 were used as a loading control. **(E)** Growth curves showing replication of DMSO-treated (control) or RAP-treated *pfpkg_2lox* parasites relative to the parental line B11. Mean values are shown from triplicate experiments. Error bars ± SD (n = 3).

parasite line called *pfpkg_2lox*. Correct integration of the *2loxPint* intron into the *pfpkg* locus of these parasites was verified by diagnostic PCR and confirmed by restriction digest analysis and Sanger sequencing of the PCR amplicon (Fig 1B and C). Western blot analysis with a polyclonal anti-PKG antibody confirmed that levels of PKG expression in *pfpkg_2lox* schizonts were indistinguishable from those of the parental B11 line (Fig 1D). To assess whether the *2loxPint* modification could affect parasite fitness or lead to undesirable gene disruption by DiCre-mediated recombination within the intron (which was not expected given the incompatibility of the *loxN* and *lox2272* sites), replication of the *pfpkg_2lox* parasites was monitored after treatment with RAP or vehicle only (DMSO control). This confirmed in both cases normal growth rates relative to the parental B11 line (Fig 1E). Collectively, these results showed that replacement of the endogenous *pfpkg* intron 3 with *2loxPint* produced no detectable defect in PKG expression or parasite growth, in turn indicating efficient splicing of the *2loxPint* intron.

### An allelic replacement approach for simultaneous disruption and complementation of the *pfpkg* gene

Having established the *pfpkg_2lox* parasite line, we next further genetically modified the parasites to enable simultaneous disruption and replacement of the *pfpkg* gene, using the paired, mutually incompatible *lox* sites within the *2loxPint* intron. To do this, we once again used a marker-free Cas9-mediated strategy to adapt the 3′ flanking sequence of the *pfpkg* locus (Fig 2A). Additional *loxN* and *lox2272* sites were introduced, positioned within intervening sequences such that DiCre-mediated recombination between the two *loxN* sites (one within the *2loxPint* and one downstream of the *pfpkg* ORF) was expected to reconstitute sequence encoding a full-length PKG fused to the fluorescent protein eGFP. In contrast, recombination between the two *lox2272* sites would instead severely truncate the endogenous PKG gene, simultaneously fusing the residual N-terminal 92 residues of the protein to mCherry. We reasoned that use of the two fluorescent reporter proteins in this way would facilitate microscopic detection of the expected recombination events after induction of DiCre activity (35, 56).

Correct integration of the 3′ targeting construct was confirmed by PCR (Fig S1A) and a clonal parasite line was obtained by limiting dilution. As expected, transient RAP treatment of this line, termed *pkg:wGFP-ckoR*, resulted in the generation of eGFP and mCherry-expressing parasites, corresponding to recombination between either the two *loxN* sites, leading to a perfect allelic replacement (PKGsynth_GFP), or between the two *lox2272* sites, resulting in disruption of the *pfpkg* gene and expression of mCherry (ΔPKG_mCherry). The expected recombination events were detectable by diagnostic PCR (Fig 2B), and expression of the predicted fusion proteins in RAP-treated parasite cultures was confirmed by Western blot using antibodies to GFP and mCherry (Fig 2C). Quantitation by flow cytometry and live microscopy of the eGFP and mCherry-positive parasites at the end of the erythrocytic cycle of RAP treatment (henceforth referred to as cycle 0) indicated that these were formed at a ratio of ~9:1 (Fig 2D), suggesting that recombination between the *loxN* sites was substantially favoured,

probably as a result of the relatively short distance between these sites in the modified *pkg:wGFP-ckoR* genome (Fig 2A). Collectively, these results confirmed that simultaneous generation of two genetically distinct populations within a single erythrocytic cycle after RAP treatment is achievable in *P. falciparum*.

### PKG has no scaffolding role during intraerythrocytic parasite development

Previous chemical genetic studies on asexual blood-stage *P. falciparum* have shown that selective inhibition of PKG activity prevents discharge of specialised secretory organelles called micronemes and exonemes, with a resulting block in egress (38). However, whether PKG expression plays a non-enzymatic role throughout the erythrocytic life cycle has not been addressed genetically.

The generation in a single step of readily distinguishable populations of PKGsynth_GFP (expected to be phenotypically wild type) and ΔPKG_mCherry (PKG-null) parasites allowed us to now examine the effects of PKG disruption or reconstitution on the entire asexual blood-stage life cycle under identical conditions. To do this, we first examined replication of control (DMSO-treated) and RAP-treated *pkg:wGFP-ckoR* parasites over the course of three erythrocytic cycles (~72 h). Because of the high proportion of PKGsynth_GFP parasites in the RAP-treated cultures, we expected to see only very small differences in overall replication rates in the cultures over this period, a prediction that was confirmed experimentally (Fig 3A). However, examination of the RAP-treated cultures revealed complete loss of the mCherry-expressing ΔPKG_mCherry parasites by the end of cycle 1, suggesting a severe growth defect upon PKG disruption (Fig 3B). To analyse this defect in more detail over the course of a single erythrocytic cycle, a highly synchronous culture containing ring-stage *pkg:wGFP-ckoR* parasites (parasitaemia ~6.5%) was RAP-treated and then incubated for a further 40 h to allow development to mature schizont stage, the point in the lifecycle at which PKG expression peaks (57). Microscopic examination over the course of this period revealed the appearance of both eGFP-expressing and mCherry-expressing mature schizonts as expected, with no decrease in total parasitaemia relative to that at the start of that cycle. This showed that all or most RAP-treated parasites were able to mature normally over the course of cycle 0, in turn indicating that ablation of PKG expression does not affect subsequent intraerythrocytic maturation within that cycle. Samples of the culture were then assessed by flow cytometry at intervals over the ensuing 3 h. As shown in Fig 3C, this revealed a gradual time-dependent decrease in the proportion of PKGsynth_GFP schizonts, presumably due to rupture of these parasites as they reached full maturity and underwent merozoite egress. In contrast, the total proportion of ΔPKG_mCherry parasites gradually increased over the 3-h period, suggesting a selective defect in rupture as mature schizonts accumulated. To further quantify this under conditions where the schizonts made up a greater proportion of the total cell population, we enriched mature cycle 0 schizonts from similar cultures and again used flow cytometry to selectively examine time-dependent changes in the proportions of fluorescent cells in the enriched schizont population. This revealed a greater than twofold increase over a 3-h period in the proportion of

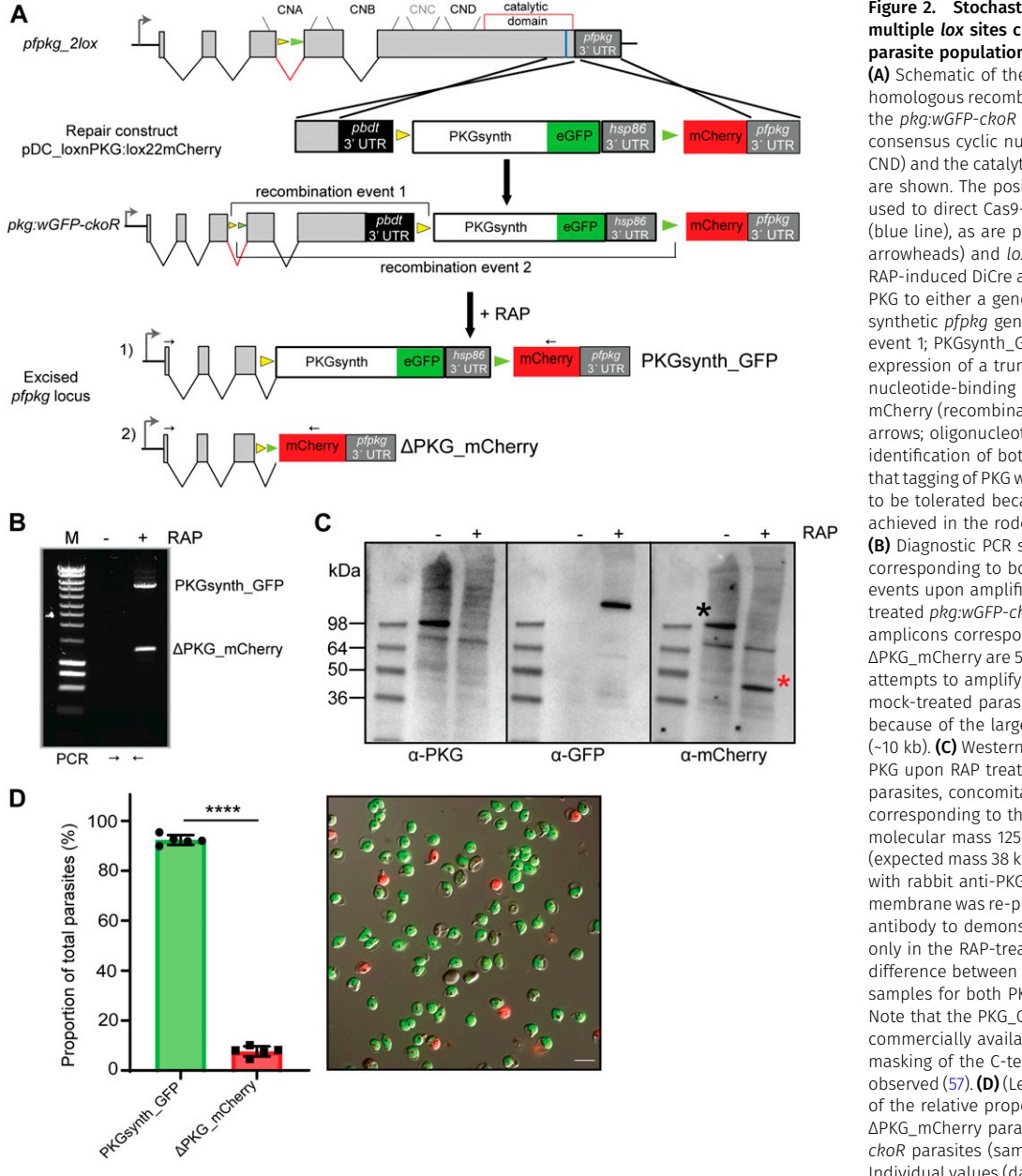

**Figure 2. Stochastic recombination between multiple *lox* sites creates genetically distinct parasite populations within a single culture.**
**(A)** Schematic of the Cas9-enhanced targeted homologous recombination approach used to create the *pkg:wGFP-ckoR* line. Positions of the four consensus cyclic nucleotide-binding domains (CNA-CND) and the catalytic domain (open red box) of PKG are shown. The position targeted by the guide RNA used to direct Cas9-mediated cleavage is indicated (blue line), as are positions of the *loxN* (yellow arrowheads) and *lox2272* (green arrowheads) sites. RAP-induced DiCre activity switches expression from wt PKG to either a gene replacement with a partially synthetic *pfpkg* gene fused to eGFP (recombination event 1; PKGsynth_GFP) or to gene disruption and expression of a truncated protein lacking the cyclic nucleotide-binding and kinase domains, fused to mCherry (recombination event 2; ΔPKG_mCherry). Black arrows; oligonucleotide primers used for identification of both events by diagnostic PCR. Note that tagging of PKG with a C-terminal eGFP was expected to be tolerated because it has previously been achieved in the rodent malaria model *P. berghei* (29). **(B)** Diagnostic PCR showing generation of products corresponding to both predicted recombination events upon amplification from genomic DNA of RAP-treated *pkg:wGFP-ckoR* parasites. Expected sizes of the amplicons corresponding to PKGsynth_GFP and ΔPKG_mCherry are 5.6 and 1.3 kb, respectively. Multiple attempts to amplify the corresponding region from mock-treated parasites (–RAP) failed, probably because of the large size of the predicted amplicon (~10 kb). **(C)** Western blot showing loss of expression of PKG upon RAP treatment of *pkg:wGFP-ckoR* parasites, concomitant with appearance of signals corresponding to the PKG_GFP fusion (expected molecular mass 125 kD) and the mCherry fusion (expected mass 38 kD; red asterisk). After being probed with rabbit anti-PKG antibodies (left-hand blot), the membrane was re-probed with a rabbit anti-mCherry antibody to demonstrate the appearance of mCherry only in the RAP-treated sample and to highlight the difference between the DMSO- and the RAP-treated samples for both PKG (black asterisk) and mCherry. Note that the PKG_GFP fusion is not recognised by the commercially available PKG antibody because of masking of the C-terminal epitope, as previously observed (57). **(D)** (Left) Quantification by flow cytometry of the relative proportions of PKGsynth_GFP and ΔPKG_mCherry parasites in RAP-treated *pkg:wGFP-ckoR* parasites (sampled at the end of cycle 0). Individual values (dark circles or squares) are shown from five biological replicate experiments (n = 5), and mean values are indicated as bars. Statistical significance was determined by unpaired *t* test (*P*-value < 0.0001). (Right) Representative image from differential inference contrast/fluorescence microscopic examination of RAP-treated *pkg:wGFP-ckoR* parasites (end of cycle 0), showing both GPF- and mCherry-positive schizonts. Scale bar, 10 μm.

ΔPKG_mCherry parasites in the schizont population (Figs 3D and S2), whereas the proportion of PKGsynth_GFP schizonts again decreased. This further supports a selective arrest in egress in the ΔPKG_mCherry mutants.

To finally examine the phenotype of PKG disruption more closely, we used time-lapse video microscopy to visualise parasite fate at egress. For this, mature schizonts enriched from RAP-treated *pkg:wGFP-ckoR* cultures were further incubated for 3 h in the presence of the reversible PKG inhibitor 4-[7-[(dimethylamino)methyl]-2-(4-fluorphenyl)imidazo[1,2-α]pyridine-3-yl]pyrimidin-2-amine (compound 2). This prevents egress while allowing schizonts to reach full

maturation, effectively synchronising the schizonts at a state of high maturation. Removal of compound 2 from wild-type parasites leads to PKG activation and schizont rupture within minutes, which can be monitored microscopically. As shown in Fig 3E and Video 1, this showed that only the PKGsynth_GFP parasites underwent egress, whereas the ΔPKG_mCherry parasites displayed no signs of parasitophorous vacuole rupture or any of the other morphological changes that generally precede egress (26, 58, 59, 60), remaining trapped inside their host red blood cells. Taken together, these results convincingly demonstrate that PKG has no detectable non-catalytic scaffolding role during maturation of asexual blood-stage

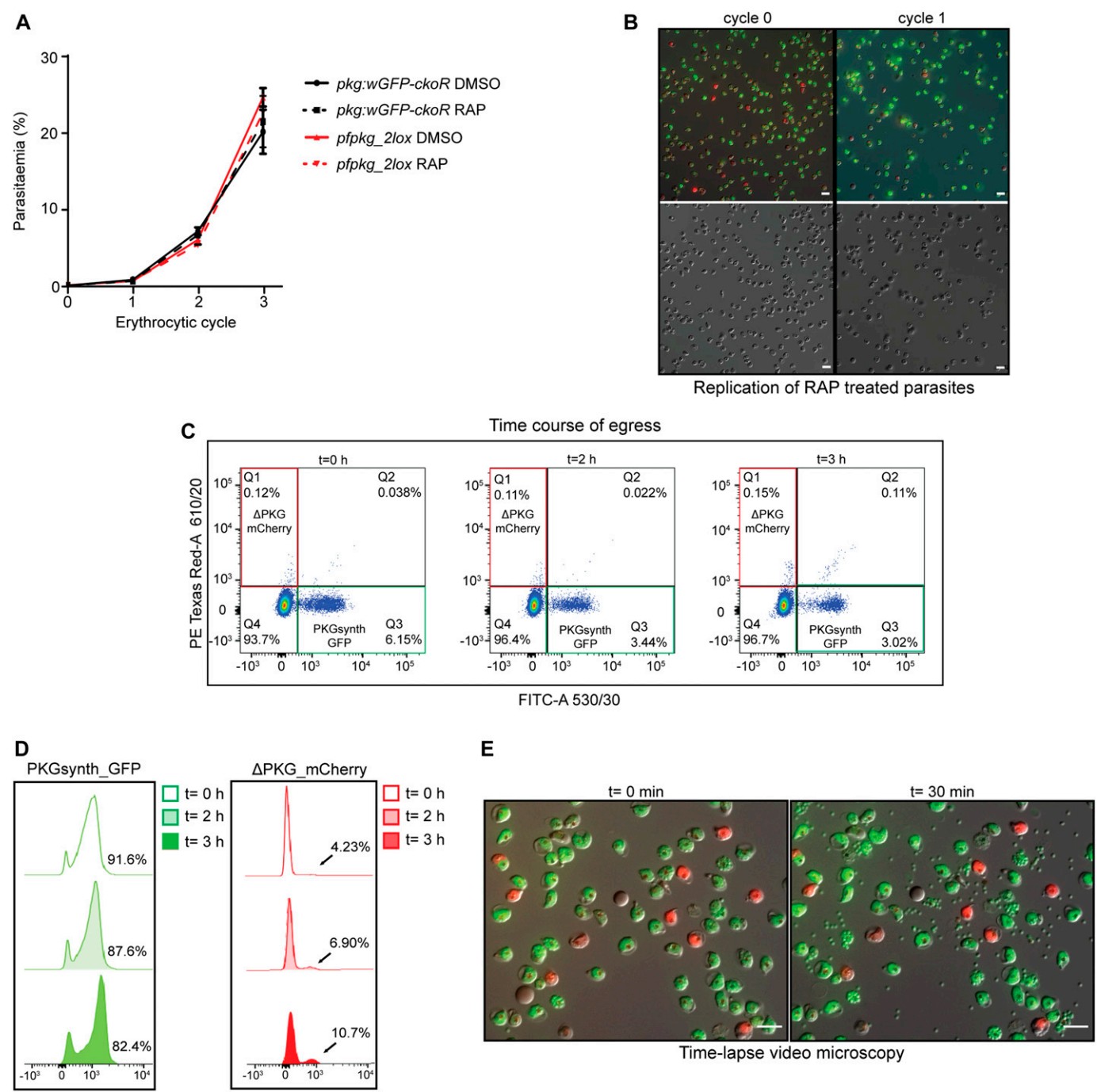

**Figure 3. PKG-null parasites undergo normal intraerythrocytic development but arrested egress.**
**(A)** Growth curves showing replication of DMSO-treated (control) or RAP-treated *pkg:wGFP-ckoR* and *pfpkg_2lox* parasites. Percentage parasitaemia values are shown (quantified by flow cytometry). Error bars ± SD (n = 3). **(B)** Differential inference contrast/fluorescence images of schizonts from a RAP-treated *pkg:wGFP-ckoR* culture, showing virtual disappearance of ΔPKG_mCherry parasites by the end of cycle 1. **(C)** Two-parameter dot plot representation of flow cytometry data monitoring the relative proportions of PKGsynth_GFP schizonts (green box, Q3; lower right-hand quadrant) and ΔPKG_mCherry schizonts (red box, Q1; upper left-hand quadrant) with time. Monitoring was initiated ~44 h after RAP treatment of a highly synchronous *pkg:wGFP-ckoRc* culture. The percentage of each population at each time point is shown within the relevant quadrant. The Q4 population predominantly represents uninfected erythrocytes. Parasitaemia at the point of RAP-treatment (the start of cycle 0) was 6.5%. **(D)** Histogram depiction of flow cytometry analysis of schizonts enriched ~44 h after RAP treatment of a *pkg:wGFP-ckoRc* culture, showing time-dependent accumulation of ΔPKG_mCherry schizonts and loss of *PKGsynth_GFP* schizonts over a 3-h time period. **(E)** Stills from time-lapse differential inference contrast/fluorescence microscopy of isolated, RAP-treated *pkg:wGFP-ckoR* schizonts after release of a compound 2–mediated egress block, showing that only the PKGsynth_GFP schizonts undergo rupture and merozoite egress. No rupture of the ΔPKG_mCherry schizonts was observed even after prolonged imaging. Scale bars, 10 μm.

schizonts, while confirming that it is essential for schizont rupture and merozoite egress.

### Removal of phosphosites renders PKG nonfunctional

Previous studies of the Cre-*lox* system in other organisms have demonstrated that the efficiency of Cre-mediated site-specific recombination generally decreases as a function of the linear "distance" between chromosomal *loxP* sites (61). In the *pkg:wGFP-ckoR* line, the genetic distance between the *loxN* and *lox2272* recombination events was 3.6 and 7.7 kb, respectively. We reasoned that this more than twofold difference was likely responsible for the relatively inefficient generation of the ΔPKG_mCherry parasites (the *lox2272* recombination event; Fig 2A). To examine whether we could obtain more similar ratios of the two recombination events, facilitating comparative analysis of the resulting genetically distinct populations, we designed a modified strategy in which recombination between the more closely situated *lox* sites would result in gene disruption, whereas recombination between the pair of more spatially distant *lox* sites would lead to allelic replacement.

To test and validate this system, we focused on the importance of the known phosphorylation sites within *P. falciparum* PKG. A total of seven phosphosites (Fig 4A) have previously been identified through several independent phosphoproteome studies by different groups (49, 50, 51, 52, 53). To examine the essentiality of phosphorylation at these sites, we decided to compare the phenotype resulting from PKG disruption with that resulting from allelic replacement with a mutant form of PKG in which the seven Ser, Thr, or Tyr residues that are targets of phosphorylation were replaced with Ala residues.

Repair construct pDC_loxNmCherry:lox22PKG (Fig S1B) was introduced into *pfpkg_2lox* parasites and, after limiting the dilution cloning, correct modification by homologous recombination of the modified *pfpkg* locus was confirmed by PCR. RAP treatment of the resulting parasite line, called *pkg:ckoR-mutGFP*, resulted in the expected recombination events as determined by diagnostic PCR (Fig 4B and C). As predicted given the more similar distances between the compatible *lox* sites in this parasite line, ratios of GFP and mCherry-positive schizonts at the end of cycle 0 were roughly comparable, demonstrating efficient generation of both ΔPKG_mCherry and PKGmut_GFP transgenic parasites in the same culture after a single RAP treatment (Fig 4D). Also as expected, the mature schizonts that appeared at the end of cycle 0 in the RAP-treated cultures were morphologically indistinguishable from those in control, DMSO-treated cultures, further indicating normal intracellular development in all the fluorescent parasites despite the fact that approximately half of the RAP-treated culture lacked expression of full-length PKG (Fig 4E), whereas the other half expressed the PKGmut_GFP mutant in which all seven phosphorylated residues were substituted with Ala residues.

To investigate the long-term viability of these parasites, RAP-treated or DMSO-treated *pkg:ckoR-mutGFP* cultures were monitored in parallel for three erythrocytic cycles. A complete arrest of parasite proliferation was observed in the RAP-treated cultures (Fig 4F). To compare the egress phenotype of the PKGmut_GFP parasites with that of the ΔPKG_mCherry parasites, we again used time-lapse microscopy to monitor egress, using as controls schizonts of the parental *pkg:ckoR-mutGFP* line. As shown in Fig 4G and Video 2, the PKGmut_GFP schizonts displayed an egress defect identical to that of the ΔPKG_mCherry parasites. To corroborate these findings, we monitored the appearance in culture supernatants of SERA5, an abundant parasitophorous vacuole protein which is released into culture supernatants upon egress (38, 62). As shown in Fig 4H, processed SERA5 was completely absent from culture supernatants of the RAP-treated *pkg:ckoR-mutGFP* schizonts. These experiments clearly showed that Ala substitution of its seven known phosphosites renders PKG functionally inactive, producing a phenotype that mimics conditional disruption of the *pfpkg* gene.

### Expanding the DiCre toolkit with the introduction of the *3loxPint* module

To unambiguously rule out a non-catalytic role for PKG during intraerythrocytic parasite growth and to expand the utility of our new toolbox for simultaneous creation of multiple allelic replacements, we examined whether it was possible to introduce a third different *lox* sequence into the *loxPint* intron to allow a third potential outcome after induction of DiCre activity. For this, we added a *lox71* site to the *2loxPint*, creating module *3loxPint* (Fig S3A and B). Cre-mediated recombination between *lox71* and the *lox66* site creates a unique *lox72*-mutant site, which is also incompatible with *loxP* (25, 63). As previously, we first precisely replaced the endogenous intron 3 of *pfpkg* in the *P. falciparum* B11 line with the *3loxPint* module. A clonal modified parasite line was obtained (called *pfpkg_3lox*) and the modification verified by diagnostic PCR, restriction digest analysis, and nucleotide sequencing (Fig S3B and C). Western blot analysis of *pfpkg_3lox* schizonts with PKG-specific antibodies showed no differences in PKG expression levels between this line, *pfpkg_2lox* and parental B11 parasites (Fig S3D), indicating correct splicing of the *3loxPint*.

To enable detection of individual recombination events, we decided as previously to design downstream modifications of the modified *pfpkg_3lox* gene such that each distinct recombination event would lead to expression of a different fluorescent protein. To validate the system and to further examine the effects of *pfpkg* disruption, we decided to design the system such that all three recombination events would lead to conditional truncation of *pfpkg* (Fig 5A), enabling us to follow maturation of these parasites under conditions in which essentially the entire culture comprised PKG-null parasites. To do this, *pfpkg_3lox* schizonts were transfected with repair construct plasmid pDC_3cKO and clonal line *pkg:3cko* obtained (Fig S1C). Synchronous ring-stage cultures of *pkg:3cko* were then treated with RAP and the resulting cycle 0 schizont-stage parasites were examined by diagnostic PCR using distinct primers designed to detect each predicted potential outcome of recombination between the various *lox* sites. All three expected recombination events were confirmed (Figs 5B and S4), supported by fluorescence microscopy examination which showed the presence of schizonts expressing mTagBFP2, eGFP, or mCherry (Fig 5C). Differential counts showed that these parasites were present in the population at proportions of 11.8% ± 1.98%, 81.6% ± 3.2%, and 6.6% ± 1.2%, respectively (n = 2), indicating a strong preference for recombination between the *loxN* sites. This latter result also confirmed correct splicing of all versions of the modified intron

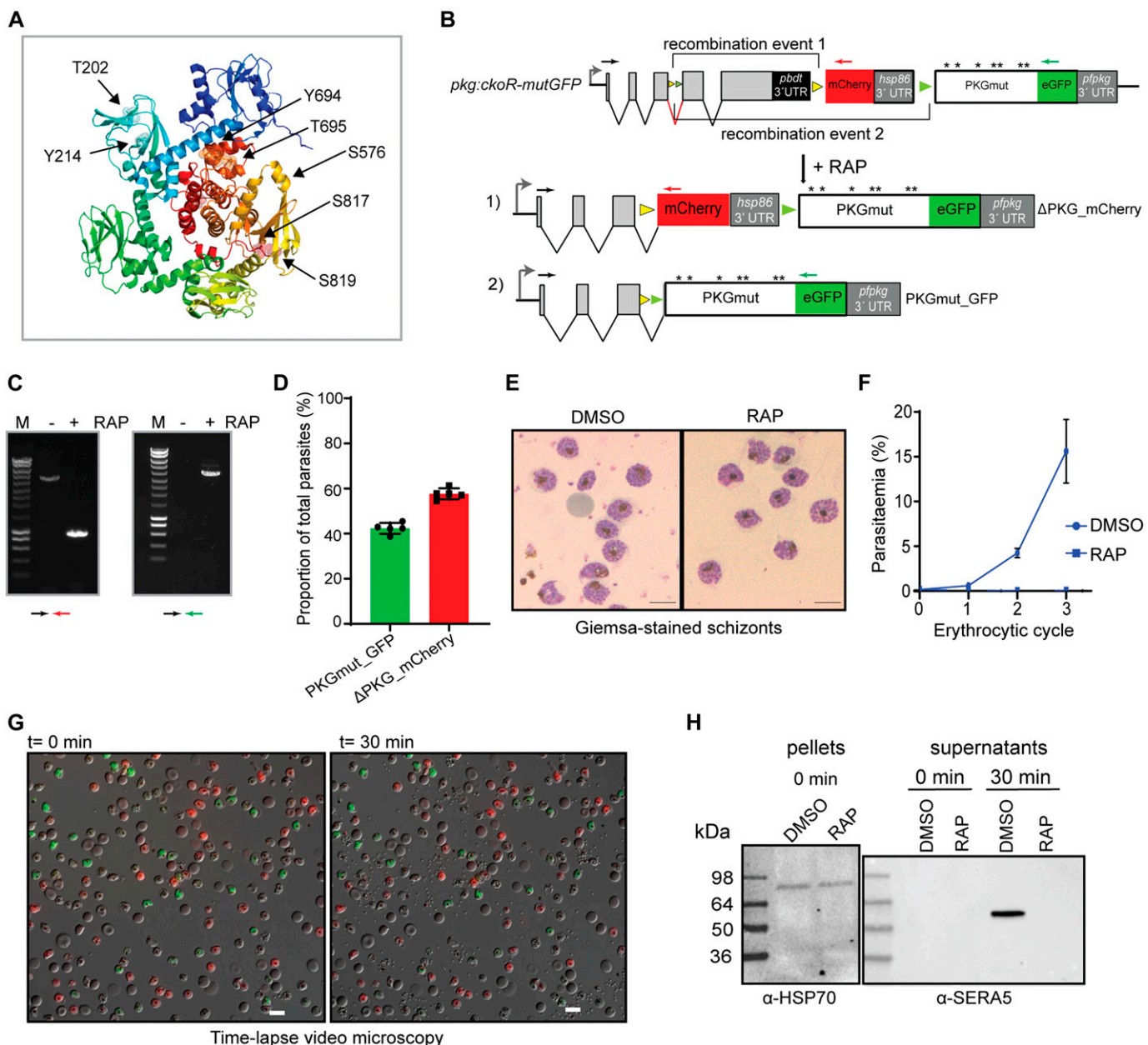

**Figure 4. Phosphosite mutations render *P. falciparum* PKG inactive.**
**(A)** Cartoon of the *P. falciparum* PKG x-ray crystal structure (PDB ID: 5DYK) in its apo form with rainbow colouring (N terminus in dark blue; C terminus in red). Cyclic nucleotide-binding domains A (dark blue), B (cyan), C (green), and D (lime) are shown, whereas the central kinase domains are in yellow/orange/red. Phosphosites identified by mass spectrometry are indicated and shown as sticks within colour-matching transparent spheres. The image was ray-traced in the PyMOL Open-Source Molecular Graphic System (https://pymol.org/2/). **(B)** Schematic of the modified *pfpkg* locus in the *pkg:cKOR-mutGFP* parasite line. Upon DiCre induction with RAP, recombination event 1 leads to conditional gene disruption (ΔPKG_mCherry), whilst recombination event 2 leads to replacement of the endogenous allele with a partially synthetic full-length allele containing Ala substitutions of all seven phosphosites (asterisks), fused to GFP (recombination event 2; PKGmut_GFP). **(C)** Diagnostic PCR results showing detection of the two distinct recombination events after DiCre activation. The amplicon specific for ΔPKG_mCherry (denoted by the black and red arrows) is ~1 kb in the RAP-treated sample and ~4.9 kb in the mock-treated (non-excised) sample. The amplicon specific for PKGmut_GFP is ~4 kb in the RAP-treated sample. Amplification from mock-treated samples was unsuccessful, likely because of the large size of the predicted fragment. **(D)** Quantification of the ratio between PKGmut_GFP and ΔPKG_mCherry schizonts in the RAP-treated parasite population at the end of cycle 0. Data shown are from five independent experiments; individual and mean values are shown. Error bars ± SD (n = 5). **(E)** Giemsa-stained images of Percoll-enriched schizonts isolated at the end of cycle 0 of DMSO- and RAP-treated *pkg:cKOR-mutGFP* parasites, showing no discernible morphological differences. Scale bar, 10 μM. **(F)** Replication of DMSO- and RAP- treated *pkg:ckoR_mutGFP* parasites over three erythrocytic cycles. Parasitaemia values shown (obtained by flow cytometry) are averages of three independent experiments. Error bars ± SD (n = 3). **(G)** Both PKGmut_GFP and ΔPKG_mCherry schizonts are defective in egress. Still images of the first and final frame of a 30-min time-lapse video of mock-treated (grey) or RAP-treated (green and red) *pkg:ckoR_mutGFP* parasites. Schizonts were synchronised by incubation with the reversible PKG inhibitor compound 2, then washed, mixed in equal proportions, and monitored for egress over a period of 30 min. Neither the red (ΔPKG_mCherry) nor the green (PKGmut_GFP) schizonts underwent egress, whereas most of the parental mock-treated *pkg:ckoR_mutGFP* schizonts ruptured. Scale bar, 10 μM. **(H)** Western blot showing that no SERA5 P50 was released into culture supernatants of RAP-treated *pkg:ckoR_mutGFP* schizonts, consistent with impaired egress in both the PKGmut_GFP and ΔPKG_mCherry schizonts.

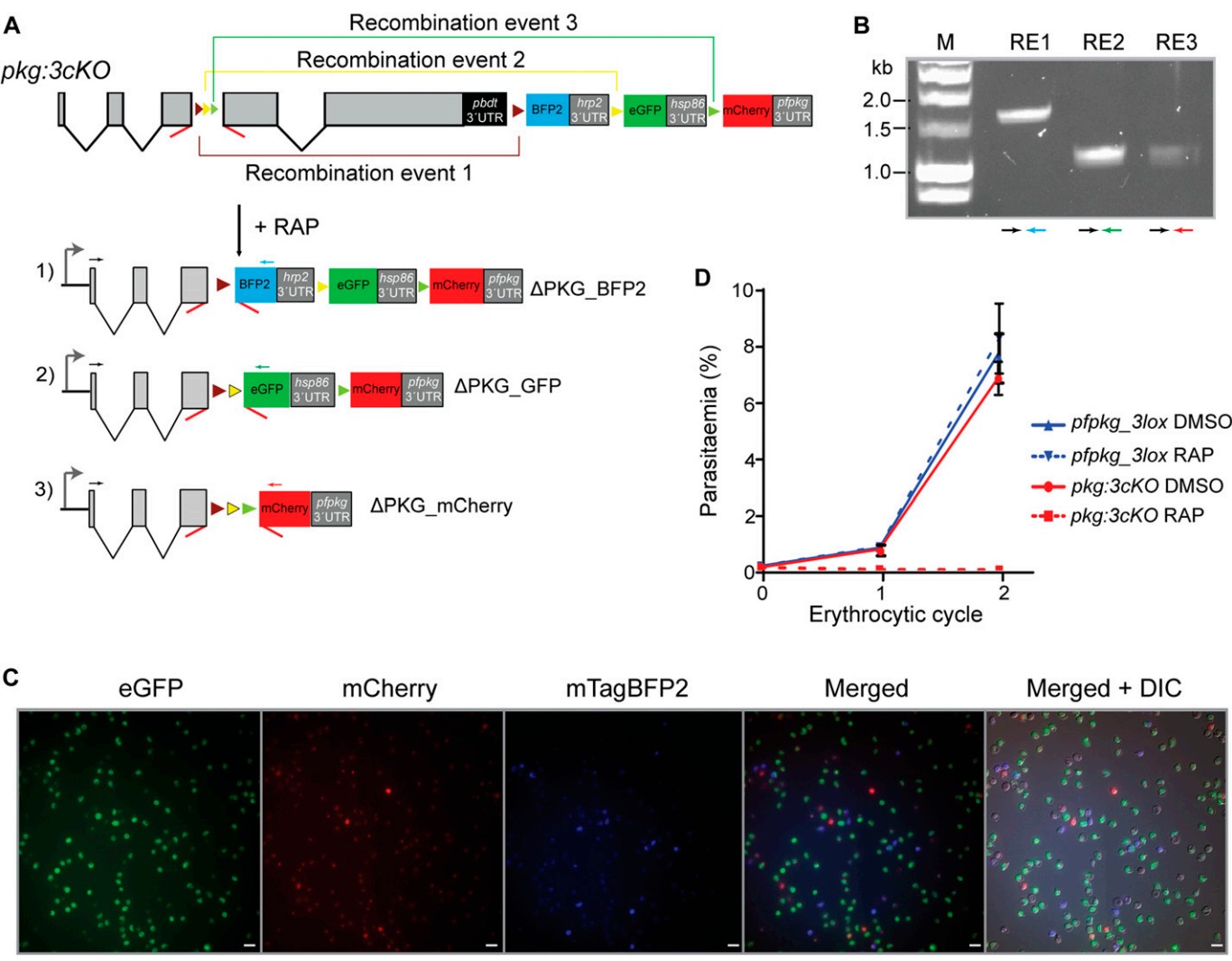

**Figure 5. Simultaneous generation of three distinct allelic exchange events.**
**(A)** Schematic of the approach used to conditionally disrupt *pfpkg* and create three distinct knockout parasite populations expressing either mTagBFP2 (ΔPKG_BFP2), eGFP (ΔPKG_GFP), or mCherry (ΔPKG_mCherry). Positions of *lox* sites are indicated with coloured arrowheads (yellow, *loxN*; green, *lox2272*; purple, *lox71*; brown, *lox66*). Positions of oligonucleotide primers used for diagnostic PCR are indicated (coloured arrows). **(B)** Confirmation by diagnostic PCR of the three recombination events (RE1, RE2, and RE3, respectively) in RAP-treated *pkg:3cKO* parasites. Coloured arrows represent identify of the primers used. **(C)** Growth assay of the *pfpkg_3lox* and *pkg:3cKO* parasite lines after mock-treatment (DMSO) or treatment with RAP. Parasitaemia was measured by flow cytometry. Error bars ± SD (n = 6). **(D)** Fluorescent microcopy images of live RAP-treated *pkg:3cKO* schizonts from the end of cycle 0, confirming the presence of all three fluorescent populations. Scale bars, 10 *μ*M.

after recombination. Importantly, there was no reduction in parasitaemia over the course of cycle 0 despite the high levels of fluorescent parasites at schizont stage, proving that disruption of PKG had no effect on intraerythrocytic parasite development. Comparison of *pfpkg_3lox* and *pkg:3cKO* parasite replication over ensuing cycles showed that RAP treatment of *pkg:3cKO* parasites led to a complete arrest in parasite growth (Fig 5D), as expected.

## Discussion

The use of site-specific recombinases and mutually incompatible *lox* sites has been a powerful tool for conditional mutagenesis and gene expression in many organisms, with a notable example being

multicolor labelling of tissues for the study of neuronal and developmental circuits in metazoa (54, 64, 65, 66). Here, we have effectively adapted this principle for use in the malaria parasite to address key challenges in *Plasmodium* genetics and to gain crucial new insights into the role of PKG in parasite blood-stage development. By introducing two or three distinct *lox* sites into an artificial *Plasmodium* intron we have been able for the first time to analyse simultaneously up to three distinct allelic replacements of PKG, allowing us to demonstrate that PKG has no essential scaffolding role during intraerythrocytic development and to genetically confirm its essentiality for egress. This is despite the fact that PKG has been shown to be expressed in asexual stages as early as 24 h post invasion, with levels of expression reaching a maximum in late schizogony (57).

Genetic complementation of *Plasmodium* knockouts with either a wild-type second copy of the gene to rescue the observed phenotype or a mutant copy to examine the role of specific residues or domains can be challenging. Episomal complementation, although technically simple and the method of choice in cases where multiple mutants need to be screened (e.g., (67)), can sometimes be partially successful because of poor segregation of plasmids in *Plasmodium*, leading to highly variable levels of expression in individual parasites and the need to maintain transgenic parasite populations under selective drug pressure (68, 69). Complementation by integration into a nonessential locus is preferable but again full restoration of the phenotype is not always possible because constitutive expression of a second gene copy can affect parasite fitness or correct trafficking of the gene product (26, 70, 71, 72). Complementation studies in animal models of malaria require extensive use of animals or the creation of mutants from independent transfections (73). The preferred method of choice is to insert a complementing allele into the authentic endogenous locus so that expression is driven by the native promoter, as described here with our new system.

The importance of genetic distance between Cre-mediated recombination events has been previously documented in other systems (61) and was also evident in our data. Experiments in ES cells and mice have revealed three important parameters in site-specific recombination: a) levels of Cre expression (which in our study should be the same in all parasites); b) the genetic distance between *lox* sites; and c) the nature of the DNA sequence (74). In parallel, studies of two distinct mutants of a gene of interest, an unequal ratio of the different recombination events is therefore to be expected, especially in the case of larger genes. This could limit the utility of the *3loxPint* system. In addition, our *3loxPint* results suggested a preference for recombination between *loxN* sites as compared with recombination between *lox2272* or *lox71*/*lox66* pairs. Multiple site–specific recombination is to our knowledge, a largely unexplored area in *Plasmodium*. It would be interesting in future experiments to study the impact of different *lox* sites on recombination ratios.

The new *loxPint* modules described here represent useful additions to the expanding toolkit (75, 76, 77, 78, 79) for conditional *Plasmodium* genetic modification, allowing up to three modifications of a gene of interest to be studied in parallel. The use of fluorescent markers combined with flow cytometry can also facilitate enrichment of the resulting parasite populations, which can be used subsequently for biochemical or phenotypical studies more readily than by generating and analysing independent lines. The system has allowed us to improve understanding of the role of PKG in blood-stage egress, providing an exciting background for further investigation of the function of this essential enzyme and its potential as a drug target.

# Materials and Methods

## Reagents and antibodies

The antifolate drug WR99210 was from Jacobus Pharmaceuticals. Rapamycin was from Sigma-Aldrich and used to treat parasites at 20 nM. The PKG inhibitor (4-[7-[(dimethylamino)methyl]-2-(4-fluorophenyl) imidazo[1,2-α]pyridine-3-yl]pyrimidin-2-amine) (compound 2) was stored at –20°C as a 10 *mM* solution in DMSO and used in cultures at 1 *μM*. For PKG detection, a rabbit polyclonal human-PKG antibody (Enzo) was used at a dilution of 1:1,000. The GFP-specific mAb 11814460001 (Roche) was used at a dilution of 1:1,000, as was a polyclonal rabbit anti-mCherry (ab167453; Abcam). A polyclonal rabbit anti-SERA5 antibody was used at 1:2,000 (80). The anti-HSP70 antibody (used at 1:1,000) was a kind gift of Dr Ellen Knuepfer, Francis Crick Institute. Restriction enzymes were from New England BioLabs and DNA ligations were performed with the Rapid DNA ligation kit (Roche).

## *P. falciparum* culture, transfection, and synchronisation

The B11 DiCre-expressing *P. falciparum* line (25) was maintained at 37°C in human erythrocytes in RPMI 1640 containing Albumax II (Thermo Fisher Scientific) supplemented with 2 mM L-glutamine and was used for all genetic modifications described. Cultures were routinely microscopically examined using Giemsa-stained thin blood films and mature schizonts were isolated by centrifugation over 70% (vol/vol) isotonic Percoll (GE Healthcare, Life Sciences) cushions. Highly synchronous ring-stage cultures were obtained by allowing schizonts to invade fresh erythrocytes for 1–2 h under shaking conditions followed by a second round of Percoll treatment and treatment of the final pellet with 5% D-sorbitol to lyse residual schizonts.

Transfections were performed as previously described (81). In brief, ~10[8] Percoll-enriched schizonts were resuspended in 100 *μl* of P3 primary cell solution (Lonza) containing 20 *μg* of Cas9 expression plasmid and 60 *μg* of linearised donor plasmid. Program FP158 of the Amaxa 4D Nucleofector X (Lonza) was used for electroporation. Drug selection with 2.5 nM WR99210 started 24 h post transfection for two cycles. Clonal lines were obtained by serial limiting dilution in flat-bottomed 96-well plates (82). Single plaques were selected and grown in the presence of 1 *μM* 5-fluorocytosine (5-FC, provided as clinical grade Ancotyl) to select for Cas9 plasmid-free and marker-free parasites.

For parasite genomic DNA extraction, the QIAGEN DNeasy Blood and Tissue kit was used. Genotype analysis diagnostic PCR was performed using Phusion polymerase (New England BioLabs).

In all cases, DiCre activity was induced by transient RAP treatment of highly synchronous early ring-stage parasites (2–3 h post invasion) as previously described (10). Parasite samples for PCR analysis of DiCre-mediated excision were collected 24 h after initiation of RAP treatment. Samples for Western blot analysis were collected at 42 h post initiation of RAP treatment.

## Plasmid construction and genotyping of transgenic lines

Sequences of the fragments used for all parasite modifications in this study are provided in Supplemental Data 1.

Parasite line *pfpkg_2lox* was created by replacing the third intron of *pfpkg* with *2loxPint* (the *P. falciparum* 3D7 *sera2* intron containing *loxN* and *lox2272* sequences, respectively). A DNA fragment was commercially obtained comprising 447 bp upstream of intron 3 as the 5′ homology arm, the *2loxPint* module, and 436 bp downstream

of intron 3 as the 3′ homology arm (GeneArt, Thermo Fisher Scientific). A single guide RNA targeting sequence TTTTAGGGTCA-TACTTTTT was inserted into a previously described pDC2 plasmid expressing Cas9, resulting in plasmid pDC2-2loxg ([19]). The repair plasmid (pMX_2lox) was linearised with BglII overnight and transfected into parasites together with plasmid pDC2_2loxg. Integration was confirmed by PCR, using primers exon1_For and exon4_Rev and restriction digest of the PCR amplicon with AccI (a full list of oligonucleotide primers used in this study is provided in Table S1). Absence of the endogenous intron 3 was confirmed by using primers exon1_For and intron3_Rev.

Construct pDC_loxnPKG:lox22mCherry was used to create line *pkg:wGFP-ckoR*. and was based on vector pDC_mCherry_MCS ([26]). The construct contains in tandem 1) a 5′ homology arm of 399 bp endogenous and 321 bp synthetic *pfpkg* sequence (obtained as a gBlock from IDT) with the PbDT 3′ UTR, 2) a fragment comprising *loxN*, the 3′ 46 bp of the *sera2* intron, a synthetic fragment of *pfpkg* starting from exon 4 (the synthetic *pfpkg* cDNA cloned in vector pTrcHis was used as a template) fused to the eGFP coding sequence ([56]) and the *pfhsp86* 3′UTR, and 3) the *lox2272* sequence the 3′ 46 bp of the *sera2* intron, the mCherry coding sequence and the *pfpkg* 3′UTR as the 3′ homology arm. A single guide RNA–targeting sequence TTGGCCGGTTAATATATCA was cloned into the Cas9 vector, generating plasmid pDC2_pkg. Vector pDC_loxnPKG:lox22mCherry was linearised overnight with ScaI and transfected together with the pDC2-pkg Cas9 plasmid into *P. falciparum* line *pfpkg_2lox*. Correct 5′ and 3′ integration was verified by PCR using primers K2_For/5int_Rev and PKGsynth_For/3int_Rev, respectively. Absence of the endogenous locus was confirmed using primers exon1_For/PKGutr_Rev.

Line *pkg:ckoR-mutGFP* was generated using repair plasmid pDC_loxNmCherry:lox22PKG. The relevant Ala substitutions were made either by overlapping PCR (for substitutions of T202A, Y214A, and S576A) using the synthetic *pfpkg* as a template, or were obtained as a gBlock fragment (Y694A, T695A, S817A, and S819A) from IDT. To create the plasmid, a synthetic fragment containing the *loxN* sequence, the 3′ 46 bp of the *sera2* intron, the *mCherry* gene, the *pfhsp86* 3′ UTR followed by *lox2272*, the 3′ 46 nucleotides of the *sera2* intron and the first 100 bp of a recodonised *pfpkg* exon 4 was commercially obtained (GeneArt, Thermo Fisher Scientific). The fragment was isolated by digest with HpaI/NcoI and cloned into a plasmid containing the rest of the mutant synthetic *pfpkg* gene fused to eGFP, followed by the *pfpkg* 3′ UTR. This intermediate vector was digested with HpaI/HindIII and the 5′ homology arm used in construct pDC_loxnPKG:lox22mCherry was cloned in, resulting in the final vector. This was linearised with ScaI overnight and transfected into line *pfpkg_2lox* with the pDC2-pkg Cas9 vector. Correct integration was confirmed as described above for line *pkg:wGFP-ckoR*.

For the *pfpkg_3lox* line, intron 3 of *pfpkg* was replaced with the *3loxPint* module (the *sera2* intron containing *lox71*, *loxN*, and *lox2272* sequences, respectively). A plasmid (pMX_3lox) containing the module spanned by the same 5′ and 3′ homology arms as used for line *pfpkg_2lox* was commercially obtained (GeneArt, Thermo Fisher Scientific). Plasmid pMX_3lox was linearised overnight with NcoI and transfected it together with plasmid pDC2_2loxg into *P. falciparum* B11 schizonts. Integration and loss of the endogenous

*pfpkg* intron 3 was confirmed by PCR and restriction digest analysis of the amplicon with AccI.

Construct pDC_3cKO was used to create line *pkg:3cKO*. Initially a plasmid, pMX_3cKO-1, containing 1) the *lox66* site followed by the 3′ 46 bp of the *sera2* intron and the mTagBFP2 coding sequence with the *pfhrp2* 3′ UTR and 2) an *loxN* site followed by the 3′ 46 bp of the *sera2* intron and the eGFP coding sequence was commercially obtained (GeneArt, Thermo Fisher Scientific). This plasmid was linearised by restriction digest with BglII. A fragment containing the hsp86 3′ UTR, the *lox2272*, the 3′ 46 bp of the *sera2* intron, *mCherry* and the *pfpkg* 3′UTR as the 3′ homology arm was isolated from plasmid pDC_loxnPKG:lox22mCherry and cloned into vector pMX_3cKO-1, resulting in construct pDC_3cKOint. This intermediate plasmid was digested with HpaI and NheI and a 3.2-kb fragment was isolated and cloned into vector pDC_loxnPKG:lox22mCherry previously digested with the same enzymes, resulting in plasmid pDC_3cKO. This was linearised overnight with ScaI and transfected together with the pDC2_pkg plasmid into *P. falciparum* line *pfpkg_3lox*.

## Parasite growth assays

To determine parasite growth rates, synchronous ring-stage parasites at 0.1% parasitaemia and 2% haematocrit were dispensed in triplicate into 12-well plates. Samples of 50 $\mu$l from each well were collected at 0, 2, 4, and 6 d, stained with SYBR Green, and analysed by flow cytometry on a BD FACSVerse using BD FACSuite software. Data were analysed using FlowJo software.

## Parasite egress assay

Parasite culture supernatants were prepared as previously described ([28]). In brief, mature schizonts were isolated by Percoll centrifugation and incubated for a further 3 h in complete medium containing compound 2 (1 $\mu$M). After removal of the inhibitor, schizonts were immediately resuspended in fresh serum-free RPMI at 37°C to allow egress. Schizont pellets and culture supernatants at t = 0 were collected as a control sample, whereas culture supernatants were collected by centrifugation after 30 min.

## Flow cytometry analysis

Parasites expressing eGFP or mCherry were quantified by flow cytometry using a FACS Aria flow cytometer (BD Biosciences). Samples were initially screened using forward and side scatter parameters and gated for erythrocytes. For eGFP detection, a 488 nm Blue Laser was used with a 530/30 filter, whereas for mCherry, a 561 nm Yellow-Green Laser was used with a 610/20 filter. For mTagBFP2 detection, the BD FACSVerse was used with a 450/50 filter.

## Immunoblotting

Synchronised schizonts were isolated by Percoll gradient centrifugation and washed in RPMI 1640 without Albumax. Parasites were extracted into a Triton X-100 buffer (20 mM Tris–HCl, pH 7.4, 150 mM NaCl, 0.5 mM EDTA, and 1% vol/vol Triton X-100, supplemented with

1× protease inhibitors [Roche]). Extracts were incubated on ice for 30 min then clarified by centrifugation at 12,000$g$ for 15 min at 4°C. Supernatants were mixed with SDS sample buffer containing DTT and incubated for 5 min at 95°C before fractionation by SDS–PAGE analysis on 4–15% Mini-PROTEAN TGX Stain-Free Protein Gels (Bio-Rad). Transfer to nitrocellulose membranes and probing for Western blot analysis was as described previously ([38]).

### Time-lapse and live fluorescence microscopy

Viewing chambers were constructed as previously described ([38]). Images were recorded on a Nikon Eclipse Ni light microscope fitted with a Hamamatsu C11440 digital camera and Nikon N Plan Apo $\lambda$ 63×/1.45NA oil immersion objective. For time-lapse video microscopy, differential inference contrast images were taken at 10-s intervals over 30 min, whereas fluorescence (GFP, mTagBFP2, and mCherry) images were taken every 2 min to prevent bleaching. Time-lapse videos were analysed and annotated using Fiji ([83]).

### Statistical analysis

All statistical analysis was carried out using GraphPad Prism 8.

## Supplementary Information

## Acknowledgements

The authors are grateful to Ellen Knuepfer for the kind gift of the $\alpha$-HSP70 antibody and to Robert Moon for sharing unpublished information on the use of mTagBFP2 in *Plasmodium*. This work was supported by Wellcome Trust grant 106239/Z/14/A (K Koussis and MJ Blackman), Wellcome Trust grant 106240/Z/14/Z (DA Baker), and Wellcome ISSF2 funding to the London School of Hygiene & Tropical Medicine. The work was also supported by funding to MJ Blackman from the Francis Crick Institute (https://www.crick.ac.uk/), which receives its core funding from Cancer Research UK (FC001043; https://www.cancerresearchuk.org), the UK Medical Research Council (FC001043; https://www.mrc.ac.uk/), and the Wellcome Trust (FC001043; https://wellcome.ac.uk/). The funders had no role in study design, data collection and analysis, decision to publish, or preparation of the manuscript.

### Author Contributions

K Koussis: conceptualization, data curation, formal analysis, methodology, and writing—original draft, review, and editing.
C Withers-Martinez: data curation and writing—original draft, review, and editing.
DA Baker: resources, supervision, funding acquisition, and writing—original draft, review, and editing.
MJ Blackman: conceptualization, resources, supervision, funding acquisition, and writing—original draft, review, and editing.

### Conflict of Interest Statement

The authors declare that they have no conflict of interest.

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
