## [Reviewer comments · Life Science Alliance]

Life Science Alliance

Simultaneous multiple allelic replacement in the malaria parasite enables dissection of PKG function

Konstantinos Koussis, Chrislaine Withers-Martinez, David Baker, and Michael Blackman

DOI: <https://doi.org/10.26508/lsa.201900626>

Corresponding author(s): Michael Blackman, The Francis Crick Institute and Michael Blackman, The Francis Crick Institute

Review Timeline:

Submission Date:	2019-12-09
Editorial Decision:	2020-01-27
Revision Received:	2020-02-29
Editorial Decision:	2020-03-02
Revision Received:	2020-03-06
Accepted:	2020-03-09

Scientific Editor: Andrea Leibfried

Transaction Report:

January 27, 2020

Re: Life Science Alliance manuscript #LSA-2019-00626-T

Prof. Michael J Blackman
The Francis Crick Institute
Division of Parasitology National Institute for Medical Research The Ridgeway
Mill Hill
London NW7 1AA

Dear Dr. Blackman,

Thank you for submitting your manuscript entitled "Simultaneous gene disruption and allelic replacement in the malaria parasite enables dissection of PKG function and phosphorylation." to Life Science Alliance. The manuscript was assessed by expert reviewers, whose comments are appended to this letter.

As you will see, the reviewers appreciate your method and the insight into PKG function provided, and they offer constructive input on how to further strengthen your manuscript. We would thus like to invite you to submit a revised version to us for publication here. Addressing the reviewer suggestions seems rather straightforward, but please do get in touch in case you would discuss a revision point further.

Thank you for this interesting contribution to Life Science Alliance. We are looking forward to receiving your revised manuscript.

Sincerely,

Andrea Leibfried, PhD

Executive Editor
Life Science Alliance
Meyerhofstr. 1
69117 Heidelberg, Germany
t +49 6221 8891 502
e a.leibfried@life-science-alliance.org
www.life-science-alliance.org

B. MANUSCRIPT ORGANIZATION AND FORMATTING:

Reviewer #1 (Comments to the Authors (Required)):

This paper from Koussis et al present an elegant manner to use the DiCre system. This is an improved version of the existing loxPint. The authors have introduced 1 or 2 supplementary loxP sites in the LoxPint, creating the 2loxPint and 3loxPint respectively. The strategy proposed allows to generate mutant/KO and complemented parasites from a single induction. One advantage of this method is the immediate availability of internal controls, since mutant and complemented

parasites are generated in parallel. As proof of principle, both 2loxPint and 3loxPint have been used on protein kinase G (PKG). In a compelling way, the authors demonstrate the involvement of PKG in the egress of *P. falciparum* merozoites. Overall, the study is very well conducted and only minor points need to be addressed.

Minor comment:

- While the efficiency of 2loxPint is well demonstrated, the efficiency of 3loxPint should be better documented. In figure 2, 2loxPint PKG_synthGFP- Δ PKG_mCherry induction generate ~90% of parasites expressing PKG_synthGFP and ~10% parasites expressing Δ PKG_mCherry. Later the authors state that the distance between the loxP sites impacts recombination efficiency. To test this, they exchange the position between Δ PKG_mCherry and PKGmut_GFP (which have the same size as PKG_synthGFP) (Figure 4). Subsequently, they observed a change in the efficiencies with 40% PKGmut_GFP and 60% Δ PKG_mCherry.

This might indicate that the addition of 5,6 kb (size of PKG_synthGFP/ PKGmut_GFP gene) before the Δ PKG_mCherry decreased recombination rates ~50% and rises some questions regarding the application limits of the 3loxPint.

This should be discussed in more detail.

-Figure 2C, what is recognised for the non-induced parasites by the mCherry antibody? Why does this band disappear upon induction with RAP?

-What is the ratio of each KO (green, red and blue) for the 3loxPint experiment (Fig.5)?

Reviewer #2 (Comments to the Authors (Required)):

In this paper the authors employ lox sites other than the previously in *P. falciparum* used loxP to 'instantly' complement conditional diCre generated mutants with different versions of a gene of interest. The authors use this to analyse the function of cGMP-dependent protein kinase (PKG). The system works in a way that a mixture of parasites is generated that harbour one of either two or three different versions of the gene of interest that are stochastically picked for complementation based on the different lox sites that are compatible only among themselves. Each version of the gene is flagged with a different fluorescent protein which makes it easy to track the parasites with a particular change in the population.

This is a very useful and clever approach that expands the toolbox of genetic techniques to study malaria parasites. The demonstration that these additional lox sites can be used in the parasite will likely also open avenues for other interesting uses. The paper also addresses a biological question and shows that PKG likely has no scaffolding function and that its 7 known phosphosites are collectively needed for its function.

The paper is very clearly and well written. The figures enable the reader to rapidly understand how the system was designed. Referencing is exemplary. Clearly this is a very nice system that will be very useful for the field. I have only some minor points:

- Figure2: PKG is expressed from 24 h after invasion. The authors conclude that there is no function of PKG before egress due to a lack of effect until parasites reach the schizont stage in cycle 0 (the cycle of excision of the functional copy of pkg). However, can it be excluded that after gene excision there is mRNA or protein left from before that could take over function? This could be easily

clarified by providing the exact timing of when the samples for the Western blots (Fig. 2C) were taken and when complete excision was detected by PCR. Was excision complete before 24 h, i.e. before transcription of the gene had started? Maybe it would be helpful to have a section in the Materials and Methods that explains how a typical rapamycin induction experiment looked (synchronisation and timing of rapamycin addition and when samples were harvested for PCR and Western blots).

- It is not always clear from the figure legends what kind of replicas were used, e.g. in Figure 1E, were these biological or technical replicas? Please add n numbers to the legends.

- the abstract states that it remains a challenge to complement conditional mutants. However, this is not always true. Episomal complementation, even of many different modifications of a gene, are possible (e.g. Mesen-Ramirez et al., PlosBio 2019). The authors rightly discuss on page 15 that such an episomal approach can be problematic and clearly the system introduced in this manuscript has the advantage of providing an 'endogenous' complementation and the benefit that this complementation happens at the same time as the knock out (e.g. avoiding problems with dominant negative effects). Nevertheless, due to its simplicity episomal complementation should not be entirely rejected. Especially if many different constructs are to be tested such an approach may be very valuable and in many cases may be less of a challenge than indicated here. The statement in the discussion could therefore be softened somewhat.

- while not essential for this paper, it still would have been nice to have the more similar ratio of recombination events between a truly complementing gene version (PFGsynth_GFP) and the knock out (reverse configuration of figure 2). It is appreciated that the authors wished to move on to test the phosphosite mutant, rather than repeating what was done in Figure 2 simply to show the corrected ratio. It would nevertheless have been nice to compare PFGsynth_GFP vs DeltaPKG_mCherry in this setup, particularly as a higher abundance of the knock out might have made it easier to analyse it.

This issue also highlights that using two different full length versions as complementation will always result in a placement of the second copy so far downstream that the ratio will be unequal. This might be a point worth mentioning for potential users and might be particularly important if large genes are to be used.

Reviewer #1 (Comments to the Authors (Required)):

*This paper from Koussis et al present an elegant manner to use the DiCre system. This is an improved version of the existing loxPint. The authors have introduced 1 or 2 supplementary loxP sites in the LoxPint, creating the 2loxPint and 3loxPint respectively. The strategy proposed allows to generate mutant/KO and complemented parasites from a single induction. One advantage of this method is the immediate availability of internal controls, since mutant and complemented parasites are generated in parallel. As proof of principle, both 2loxPint and 3loxPint have been used on protein kinase G (PKG). In a compelling way, the authors demonstrate the involvement of PKG in the egress of *P. falciparum* merozoites. Overall, the study is very well conducted and only minor points need to be addressed.*

Minor comment:

- While the efficiency of 2loxPint is well demonstrated, the efficiency of 3loxPint should be better documented. In figure 2, 2loxPint PKG_{synthGFP}-ΔPKG_{mCherry} induction generate ~90% of parasites expressing PKG_{synthGFP} and ~10% parasites expressing ΔPKG_{mCherry}.

Later the authors state that the distance between the loxP sites impacts recombination efficiency. To test this, they exchange the position between ΔPKG_{mCherry} and PKGmut_{GFP} (which have the same size as PKG_{synthGFP}) (Figure 4). Subsequently, they observed a change in the efficiencies with 40% PKGmut_{GFP} and 60% ΔPKG_{mCherry}.

This might indicate that the addition of 5,6 kb (size of PKG_{synthGFP}/ PKGmut_{GFP} gene) before the ΔPKG_{mCherry} decreased recombination rates ~50% and rises some questions regarding the application limits of the 3loxPint. This should be discussed in more detail.

Our aims in this part of the work were: (a) to show that recombination can be achieved between 3 distinct loxP sites within a synthetic intron; and (b) to use this tool to exclude any role for PKG in intracellular development of the parasite prior to the point of egress. We fully agree with the reviewer that the genetic distance between introduced loxP sites will likely be a limiting factor for the broader utility of the 3loxPint strategy. Despite these limitations however, we believe that the 3loxPint module may prove particularly useful for the allelic replacement of small genes or for other applications such as modification of promoter regions or using a combination of excision and inversion to achieve multiple simultaneous modifications. As requested by the reviewer, in the revised manuscript the potential limitations and uses of the 3loxPint system are now discussed further in the Discussion (lines 371-379).

-Figure 2C, what is recognised for the non-induced parasites by the mCherry antibody? Why does this band disappear upon induction with RAP?

The anti-mCherry panel in Fig 2C (right hand-side panel) shows a re-probing of the PKG Panel (the left hand-side panel). After being probed with rabbit anti-PKG antibodies, the membrane was re-probed with a rabbit anti-mCherry antibody to demonstrate the appearance of mCherry only in the RAP-treated sample and to highlight the difference between the DMSO- and the RAP-treated samples for both PKG and mCherry. To clarify this, in the revised manuscript the PKG band is now labelled in Fig 2C (black asterisk) and referred to in the figure legend.

-What is the ratio of each KO (green, red and blue) for the 3loxPint experiment (Fig.5)?

We apologise for the omission of these data from the original manuscript. The proportions of each population of fluorescent parasites in the RAP-treated cultures are now included in the revised text (lines 332-334). The apparent preference for recombination between *loxN* sites is also discussed (lines 379-383).

Reviewer #2 (Comments to the Authors (Required)):

In this paper the authors employ lox sites other than the previously in P. falciparum used loxP to 'instantly' complement conditional diCre generated mutants with different versions of a gene of interest. The authors use this to analyse the function of cGMP-dependent protein kinase (PKG). The system works in a way that a mixture of parasites is generated that harbour one of either two or three different versions of the gene of interest that are stochastically picked for complementation based on the different lox sites that are compatible only among themselves. Each version of the gene is flagged with a different fluorescent protein which makes it easy to track the parasites with a particular change in the population.

This is a very useful and clever approach that expands the toolbox of genetic techniques to study malaria parasites. The demonstration that these additional lox sites can be used in the parasite will likely also open avenues for other interesting uses. The paper also addresses a biological question and shows that PKG likely has no scaffolding function and that its 7 known phosphosites are collectively needed for its function.

The paper is very clearly and well written. The figures enable the reader to rapidly understand how the system

was designed. Referencing is exemplary. Clearly this is a very nice system that will be very useful for the field. I have only some minor points:

- *Figure 2: PKG is expressed from 24 h after invasion. The authors conclude that there is no function of PKG before egress due to a lack of effect until parasites reach the schizont stage in cycle 0 (the cycle of excision of the functional copy of pkg). However, can it be excluded that after gene excision there is mRNA or protein left from before that could take over function? This could be easily clarified by providing the exact timing of when the samples for the Western blots (Fig. 2C) were taken and when complete excision was detected by PCR. Was excision complete before 24 h, i.e. before transcription of the gene had started? Maybe it would be helpful to have a section in the Materials and Methods that explains how a typical rapamycin induction experiment looked (synchronisation and timing of rapamycin addition and when samples were harvested for PCR and Western blots).*

In all our experiments using the DiCre system, we RAP-treated highly synchronised newly invaded ring-stage parasites (3-4 hours post invasion). Samples of the culture for analysis by diagnostic PCR to assess excision efficiency were collected 24 h following RAP treatment (at trophozoite stage) whilst samples for Western blots were collected 40-42 h post RAP-treatment. These important details are now included in the Material and Methods section of the revised manuscript (lines 431-435).

With regard to the point raised by the reviewer, previous studies examining PKG expression over the course of the asexual blood stage cycle (Hopp et al, 2012) showed that PKG protein can be detected from 24 h post-invasion onwards. In our current study, although we cannot exclude the presence of traces of PKG mRNA at 24 h post invasion in our knockout parasites, we believe that the key point is that we could not detect protein signal at 44 h in the majority of the treated parasites, suggesting efficient depletion of endogenous PKG expression in cycle 0. Whilst it could be argued that undetectably low levels of enzyme present in RAP-treated cells might have been sufficient to fulfil an essential role for PKG prior to egress, they were clearly not sufficient to initiate the egress cascade at the end of cycle 0. We therefore maintain that the simplest interpretation of our data is that PKG does not play an essential role during intraerythrocytic development of the parasite.

- *It is not always clear from the figure legends what kind of replicas were used, e.g. in Figure 1E, were these biological or technical replicas? Please add n numbers to the legends.*

Reference to the number of biological replicates in each case have now been added now to the figure legends.

- the abstract states that it remains a challenge to complement conditional mutants. However, this is not always true. Episomal complementation, even of many different modifications of a gene, are possible (e.g. Mesen-Ramirez et al., PlosBio 2019). The authors rightly discuss on page 15 that such an episomal approach can be problematic and clearly the system introduced in this manuscript has the advantage of providing an 'endogenous' complementation and the benefit that this complementation happens at the same time as the knock out (e.g. avoiding problems with dominant negative effects). Nevertheless, due to its simplicity episomal complementation should not be entirely rejected. Especially if many different constructs are to be tested such an approach may be very valuable and in many cases may be less of a challenge than indicated here. The statement in the discussion could therefore be softened somewhat.

We completely agree with the reviewer that genetic complementation using episomal (plasmid) expression constructs can be a perfectly adequate and occasionally even preferred means of assessing gene function in *P. falciparum*, especially where analysis of multiple different mutant constructs is under investigation. Indeed, we have ourselves used episomal approaches on several previous occasions in our own studies. As the reviewer correctly points out, what we provide here is an alternative experimental strategy to overcome the problems associated with either constitutive expression of a second gene copy *prior* to endogenous gene disruption (which can be toxic), or the variability in gene expression often associated with episomal constructs, which are generally poorly segregated in *Plasmodium*. In response to the reviewer's suggestions, the revised Discussion has now been modified to clarify these issues (lines 356-360).

- while not essential for this paper, it still would have been nice to have the more similar ratio of recombination events between a truly complementing gene version (PFGsynth_GFP) and the knock out (reverse configuration of figure 2). It is appreciated that the authors wished to move on to test the phosphosite mutant, rather than repeating what was done in Figure 2 simply to show the corrected ratio. It would nevertheless have been nice to compare PFGsynth_GFP vs DeltaPKG_mCherry in this setup, particularly as a higher abundance of the knock out might have made it easier to analyse it.

This issue also highlights that using two different full length versions as complementation will always result in a placement of the second copy so far downstream that the ratio will be unequal. This might be a point worth mentioning for potential users and might be particularly important if large genes are to be used.

We appreciate the reviewer's comments and fully agree that the control line described could have been generated and included in the analysis; as the reviewer mentions this would facilitate analysis of the knockout phenotype. The positioning of the recombination events is important and needs to be considered in experimental

design using the 2loxPint or 3 loxPint modules. This is now mentioned in the revised Discussion. We also agree that there are limitations to the system especially in the case of larger genes, and so this issue is now also discussed further (see also our response to the first comment of Reviewer #1).

March 2, 2020

RE: Life Science Alliance Manuscript #LSA-2019-00626-TR

Prof. Michael J Blackman
The Francis Crick Institute
Division of Parasitology National Institute for Medical Research The Ridgeway
Mill Hill
London NW7 1AA

Dear Dr. Blackman,

Thank you for submitting your revised manuscript entitled "Simultaneous multiple allelic replacement in the malaria parasite enables dissection of PKG function". I have now assessed the revisions performed and I appreciate the introduced changes. I would thus be happy to publish your paper in Life Science Alliance pending final revisions necessary to meet our formatting guidelines:

- Please link your profile in our submission system to your ORCID iD, you should have received an email with instructions on how to do so
- Please re-name appendix Table S1 to simply supplementary Table S1 and provide it as a docx file
- Please provide the integration constructs used (current appendix figure S1) as a dataset and add a callout to this dataset in the manuscript text
- Figure S5 is missing from the submission, please provide
- Please add legends for the movies
- Please add a scale bar to Figure 4E and increase visibility of those in Fig. 3B, Fig. 4G. Fig. 5C

A. FINAL FILES:

B. MANUSCRIPT ORGANIZATION AND FORMATTING:

Sincerely,

March 9, 2020

RE: Life Science Alliance Manuscript #LSA-2019-00626-TRR

Prof. Michael J Blackman
The Francis Crick Institute
Malaria Biochemistry Laboratory
1 Midland Road
London NW1 1AT
United Kingdom

Dear Dr. Blackman,

Thank you for submitting your Methods entitled "Simultaneous multiple allelic replacement in the malaria parasite enables dissection of PKG function". It is a pleasure to let you know that your manuscript is now accepted for publication in Life Science Alliance. Congratulations on this interesting work.

DISTRIBUTION OF MATERIALS:

Again, congratulations on a very nice paper. I hope you found the review process to be constructive and are pleased with how the manuscript was handled editorially. We look forward to future exciting submissions from your lab.

Sincerely,

Andrea Leibfried, PhD
Executive Editor
Life Science Alliance
Meyerohofstr. 1
69117 Heidelberg, Germany
t +49 6221 8891 502
e a.leibfried@life-science-alliance.org
www.life-science-alliance.org